# Gastric Cancer and Intestinal Metaplasia: Differential Metabolic Landscapes and New Pathways to Diagnosis

**DOI:** 10.3390/ijms25179509

**Published:** 2024-09-01

**Authors:** Seong Ji Choi, Hyuk Soon Choi, Hyunil Kim, Jae Min Lee, Seung Han Kim, Jai Hoon Yoon, Bora Keum, Hyo Jung Kim, Hoon Jai Chun, Youngja H. Park

**Affiliations:** 1Department of Internal Medicine, Hanyang University College of Medicine, Seoul 04763, Republic of Korea; drcoolandy@gmail.com (S.J.C.); jaihoonyoon@hanyang.ac.kr (J.H.Y.); 2Division of Gastroenterology and Hepatology, Department of Internal Medicine, Korea University College of Medicine, Seoul 02841, Republic of Korea; mdkorea@gmail.com (H.S.C.); jmlee1202@gmail.com (J.M.L.); kimseunghan09@gmail.com (S.H.K.); borakeum@hanmail.net (B.K.); hjkimmd@korea.ac.kr (H.J.K.); 3EN BIO, Cheongju-si 28494, Republic of Korea; kimhyunil7071@gmail.com

**Keywords:** gastric cancer, intestinal metaplasia, metabolomics, metabolite profiling, biomarker

## Abstract

Gastric cancer (GC) is the fifth most common cause of cancer-related death worldwide. Early detection is crucial for improving survival rates and treatment outcomes. However, accurate GC-specific biomarkers remain unknown. This study aimed to identify the metabolic differences between intestinal metaplasia (IM) and GC to determine the pathways involved in GC. A metabolic analysis of IM and tissue samples from 37 patients with GC was conducted using ultra-performance liquid chromatography with tandem mass spectrometry. Overall, 665 and 278 significant features were identified in the aqueous and 278 organic phases, respectively, using false discovery rate analysis, which controls the expected proportion of false positives among the significant results. sPLS-DA revealed a clear separation between IM and GC samples. Steroid hormone biosynthesis, tryptophan metabolism, purine metabolism, and arginine and proline metabolism were the most significantly altered pathways. The intensity of 11 metabolites, including N1, N2-diacetylspermine, creatine riboside, and N-formylkynurenine, showed significant elevation in more advanced GC. Based on pathway enrichment analysis and cancer stage-specific alterations, we identified six potential candidates as diagnostic biomarkers: aldosterone, N-formylkynurenine, guanosine triphosphate, arginine, S-adenosylmethioninamine, and creatine riboside. These metabolic differences between IM and GC provide valuable insights into gastric carcinogenesis. Further validation is needed to develop noninvasive diagnostic tools and targeted therapies to improve the outcomes of patients with GC.

## 1. Introduction

Gastric cancer (GC) is the fifth most common cancer type and the fifth leading cause of cancer-related deaths globally [1]. In the Republic of Korea, GC imposes a significant socioeconomic burden, particularly among individuals who do not participate in the national cancer screening program [2]. GC is more common in men; in developed countries, men are 2.2 times more likely to be diagnosed with GC than women. Owing to the asymptomatic nature of the disease, patients are often diagnosed at advanced stages when the disease becomes inoperable, either surgically or endoscopically. The 5-year survival rate varies widely depending on the cancer stage at diagnosis, ranging from less than 20% in late stages to >90% in early stages [3]. Endoscopy is currently the most reliable and widely used method for diagnosing GC. However, it has several disadvantages, including invasiveness, significant miss rates, risk of infection, and complications [4,5]. Miss rates are particularly high in patients with small-sized or non-elevated GC morphology [6]. Therefore, the early detection and prediction of GC are crucial for improving outcomes and preventing disease.

To address these challenges, various studies have focused on identifying effective diagnostic methods and developing novel biomarkers. Carbohydrate antigen family members (72-4, 19-9, and 125) and carcinoembryonic antigen (CEA) have been used as serum-based biomarkers for GC detection [7,8]. However, these markers are not exclusive to GC and are altered in other types of cancers [7,9]. The development of GC-specific biomarkers requires high disease specificity. Proteomic [10] and genomic approaches [11] have also been explored to identify GC; however, metabolites provide the most accurate presentation of the molecular phenotype. Metabolite-associated biomarkers have been identified in various cancer types such as breast, brain, ovarian, and prostate cancers and are currently used for diagnosis and prognosis prediction [12,13,14,15]. Prognostic biomarkers, including the neutrophil-to-lymphocyte ratio, have also been proposed, particularly in patients with advanced GC treated with Programmed Cell Death Protein 1 (PD-1) and Programmed Death-Ligand 1 (PD-L1) agents [16]. In this study, we aimed to identify diagnostic biomarkers specific to GC using metabolomics.

It is widely accepted that GC develops through a multistep cascade: gastritis, atrophy, intestinal metaplasia (IM), dysplasia, and cancer [17]. As IM is considered the most significant stage in GC development [18], the phenotypic transition from IM to GC is a key step in gastric carcinogenesis. Previous studies have applied metabolomics to assess patients with GC using gastric tissue, blood, urine, or gastric juice samples [19]. Among the studies using tissues, most compared GC tissue metabolites with those from normal tissue [20,21,22], non-cancerous [23], and chronic superficial gastritis tissues [24].

In this study, we aimed to identify the metabolic landscapes of IM and GC and discover diagnostic biomarkers for the progression from IM to GC by comparing their metabolic differences.

## 2. Results

### 2.1. Patient Characteristics

We enrolled 37 patients and collected two GC tissue samples and two corresponding IM tissue samples from each patient, investigating the aqueous and organic metabolites obtained from these tissues (Figure 1a). The cancer location, stage, pathological differentiation, and Lauren classification were used to obtain detailed information on the participants. Based on the American Joint Committee on Cancer Tumor-Node-Metastasis (TNM) system, 15 patients with stage I, 7 with stage II, 4 with stage III, and 11 with stage IV were included in this study. The history of *Helicobacter pylori* infection and family history of GC were also considered. Table 1 shows the characteristics of patients with GC.

### 2.2. Analytical Performance Evaluation

A schematic overview of the biomarker identification criteria is shown in Figure 1b. Data were extracted using adaptive processing of liquid chromatography-mass spectrometry and subjected to the xMSanalyzer. To assess the performance of the analytical system, the processed data from the xMSanalyzer (24,591 features in the aqueous phase and 18,980 features in the organic phase) were input into xmsPANDA version 1.3.2. A total of 665 significant features from the aqueous phase and 278 features from the organic phase were selected based on a false discovery rate FDR q-value of ≤0.05 using the Limma R package. With significant features identified in the FDR analysis (q ≤ 0.05), principal component analysis (PCA) showed a clear separation between case samples (yellow and red) and pool samples (green), in both aqueous and organic phases (Figure 2).

### 2.3. Uni/Multivariate Observation of Metabolic Alterations between IM and GC

#### 2.3.1. Comparison of Aqueous Data

For a detailed investigation of metabolic differences caused by GC, the aqueous extracted data from the IM and GC groups were analyzed using Manhattan plot (FDR q ≤ 0.05), principal component analysis (PCA), and sparse partial least squares discriminant analysis (sPLS-DA). A total of 665 features (FDR q ≤ 0.05)) were selected as discriminatory features from the aqueous extract data using a *t*-test, and presented as colored dots in the Manhattan plot (Figure 3a). −log_10_(*p*) values are shown on the y-axis, whereas the mass/charge ratio (*m*/*z*) are displayed on the x-axis (Figure 3a).

The dashed line (FDR q ≤ 0.05) separates the significant metabolites from the input features. Above the line, the blue dots represent the features that are highly expressed in GC, while the red dots represent the features that are poorly expressed in GC. Using 665 features (FDR q ≤ 0.05), hierarchical cluster analysis (HCA) did not clearly separate the two groups (Figure 3b). Therefore, to obtain clear separation, sPLS-DA was applied, which resulted in better separation compared with PCA (Figure 3c).

#### 2.3.2. Comparison of Organic Data

Following the analyses of the aqueous data, a Manhattan plot, PCA, and sPLS-DA were employed to assess the metabolic differences in the organic data. In total, 278 features (FDR q ≤ 0.05), shown as colored dots in the Manhattan plot, were selected as discriminatory features from the organic extract data (Figure 3d). An explanation of the Manhattan plot was provided in the previous section. Using 278 features (FDR q ≤ 0.05), the two groups were not clearly separated by HCA (Figure 3e). Therefore, to obtain clear separation, sPLS-DA was applied, which resulted in better separation compared with PCA (Figure 3f).

### 2.4. Pathway Enrichment Analysis between IM and GC

The 665 and 278 significant features (FDR q ≤ 0.05) identified from the comparison of aqueous data and organic data, respectively, were matched to the Kyoto Encyclopedia of Genes and Genomes (KEGG) IDs using xMSannotator version 1.3.2. The matched KEGG IDs that coped with the intensity table were input into the Metaboanalyst 5.0 (www.metaboanalyst.ca (accessed on 12 April 2024)) for pathway analysis. As a result, highly affected pathways, including pathway name, −log_10_(*p*), pathway hits, and FDR, were determined (Figure 4), including steroid hormone biosynthesis [−log_10_(*p*) = 8.13, impact = 0.24], tryptophan metabolism [−log_10_(*p*) = 2.27, impact = 0.38], purine metabolism [−log_10_(*p*) = 1.56, impact = 0.21], and arginine and proline metabolism [−log_10_(*p*) = 0.75, impact = 0.24].

### 2.5. Relative Intensities of Metabolites Significantly Altered by GC

We used molecular tolerance within the range of 10 ppm to annotate the ion features, and fragmented the ion features to validate them with standard chemicals. Several significantly altered metabolites were identified in the significant pathways. In steroid hormone biosynthesis, metabolite alterations start with two metabolites: 21-deoxycortisol (*m*/*z*: 347.2185, [M+H]^+^) and androst-4-ene-3,17-dione (Figure 5a and Figure 5b, respectively). The levels of 21-deoxycortisol and metabolites derived from 21-deoxycortisol, including corticosterone (*m*/*z*: 347.2185, [M+H]^+^), aldosterone (*m*/*z*: 361.2033, [M+K]^+^), 11β, 21 dihydroxy-5β-pregnane-3,20-dione (*m*/*z*: 387.1921, [M+K]^+^), and 21-dihydroxy-5β-pregnane-3,11,20-trione (*m*/*z*: 347.2185, [M+H]^+^) were significantly lower in GC than in IM. The levels of metabolites derived from androst-4-ene-3,17-dione, including 11β-hydroxyandrost-4-ene-3,17-dione (*m*/*z*: 267.1751, [M+H-2H_2_O]^+^), adrenosterol (*m*/*z*: 283.1713, [M+H-H_2_O]^+^), etiocholan-3α-ol-17-one (*m*/*z*: 329.1866, [M+K]^+^), 19-hydroxyandrost-4-ene-3,17-dione (*m*/*z*: 267.1751, [M+H-2H_2_O]^+^), 19-oxoandrost-4-ene-3,17-dione (*m*/*z*: 283.1713, [M+H-2H_2_O]^+^), and androsterone (*m*/*z*: 329.1866, [M+K]^+^), were significantly altered by GC. Among these, the levels of adrenosterone and 19-oxoandrost-4-ene-3,17-dione were significantly higher in the GC group than in the IM group, while those of 11β, 21-dihydroxy-5β-pregnane-3,20-dione, etiocholan-3α-ol-17-one, 19-hydroxyandrost-4-ene-3,17-dione, and androsterone were significantly lower in the GC group than in the IM group.

In tryptophan metabolism (Figure 6), the levels of L-tryptophan (*m*/*z*: 205.0964, [M+H]^+^) and its metabolic derivatives, including indole (*m*/*z*: 188.0702, [M+H]^+^), indole acetaldehyde (*m*/*z*: 160.0750, [M+H]^+^), N-formyl kynurenine (*m*/*z*: 254.1141, [M+NH_4_]^+^), indoleacetate (*m*/*z*: 170.0600, [M+H]^+^), and L-kynurenine (*m*/*z*: 191.0820, [M+H]^+^) were significantly altered. N-formylkynurenine was significantly higher in the GC group than in the IM group, while those of indole, L-tryptophan, indole acetaldehyde, indoleacetate, and L-kynurenine were significantly lower in the GC group than in the IM group.

In purine metabolism (Figure 7), the expression levels of guanosine 3′,5′-bis(diphosphate) (*m*/*z*: 437.3722, [M+Na]^+^) and its metabolic derivatives (including guanosine triphosphate (GTP) (*m*/*z*: 523.9958, [M+H]^+^), inosine (*m*/*z*: 307.0433, [M+K]^+^), hypoxanthine (*m*/*z*: 119.0350, [M+H-H_tyl_0]^+^), guanosine (*m*/*z*: 322.0538, [M+K]^+^), and guanine (*m*/*z*: 152.0580, [M+H]^+^)) and 5′-phosphoribosyl-N-formylglycinamide (*m*/*z*: 279.0390, [M+H-2H_2_0]^+^) and its metabolic derivatives (including 2-(formamido)-N1-(5′-phosphoribosyl)acetamidine (*m*/*z*: 296.0659, [M+H-H_2_0]^+^) and aminoimidazole ribotide (*m*/*z*: 296.0659, [M+H]^+^)) were significantly altered. The expression level of GTP was significantly higher in the GC group than in the IM group, while those of guanosine 3′,5′-bis(diphosphate), inosine, hypoxanthine, guanosine, guanine, 5′-Phosphoribosyl-N-formylglycinamide, 2-(formamido)-N1-(5′-phosphoribosyl)acetamidine, and aminoimidazole ribotide were significantly lower in the GC group than in the IM group.

In arginine and proline metabolism (Figure 8), the expression levels of S-adenosylmethioninamine (*m*/*z*: 150.0865, [M+Na]^+^), N-acetylputrescine (*m*/*z*: 113.1071, [M+H-H_2_O]^+^), N_4_-acetylaminobutanal (*m*/*z*: 152.0694, [M+Na]^+^), and N-carbamoyl sarcosine (*m*/*z*: 150.0865, [M+NH_4_]^+^) were significantly higher in the GC group than in the IM group, while those of arginine (*m*/*z*: 175.1194, [M+H]^+^), creatine (*m*/*z*: 132.0766, [M+H]^+^), sarcosine (*m*/*z*: 90.0543, [M+H]^+^), N-methylhydantoin (*m*/*z*:, [M+NH_4_]^+^), and creatinine (*m*/*z*: 136.0467, [M+Na]^+^) were significantly lower in GC than in the GC group than in the IM group.

### 2.6. Alteration of Metabolite Intensities According to GC Stage

Figure 9 shows the significantly altered metabolite intensities according to stage. The significance of all metabolites was calculated by the analysis of variance (ANOVA) with a Tukey post hoc test (*p* ≤ 0.05), using GraphPad Prism. The expression levels of 11 metabolites, including N1, N2-diacetylspermine (*m*/*z*: 287.2430, [M+H]^+^), creatine riboside (*m*/*z*: 281.1446, [M+NH_4_]^+^), (3z,6z)-3,6-nonadienal (*m*/*z*: 161.0922, [M+H]^+^), N-formyl kynurenine (*m*/*z*: 237.0870, [M+H]^+^), S-adenosylmethioninamine (*m*/*z*: 339.1629, [M+H-H_2_O]^+^), methionyl-leucine (*m*/*z*: 263.1399, [M+H]^+^), norepinephrine (*m*/*z*: 187.1073, [M+NH_4_]^+^), S-(formylmethyl)glutathione (*m*/*z*: 357.1258, [M+NH_4_]^+^), methionylphenylalanine (*m*/*z*: 297.1287, [M+H]^+^), oxindole-3-acetate (*m*/*z*: 192.0654, [M+H]^+^), and N1-acetylspermine (*m*/*z*: 230.0954, [M+H]^+^), significantly increased according to stage.

## 3. Discussion

Most patients with GC are diagnosed at an advanced stage, resulting in poor prognosis and limited treatment options [1,25]. Current biomarkers for GC diagnosis and prognosis have low sensitivity and specificity [26]. Hence, most of the current diagnoses are based on invasive endoscopy. Thus, less invasive diagnostic tools and more specific biomarkers must be developed for the early detection of GC [27]. This study aimed to identify the metabolic alterations caused by GC. PLS-DA showed significant differences in the metabolic phenotypes during the progression of GC. Our analysis identified key metabolites, including aldosterone, N-formylkynurenine, GTP, arginine, and creatine riboside, which exhibited distinct alterations from high-impact pathways or stage-dependent analysis. These metabolites have potential as diagnostic biomarkers for GC, serving as candidates for the identification and prediction of the disease. To the best of our knowledge, this is the first metabolomic study to differentiate between GC and IM using tissue samples.

In this study, we extracted 943 significant metabolites from both aqueous and organic phases. Through pathway enrichment analysis and ANOVA tests with cancer stages, we aimed to identify potential candidates for discriminating GC from IM samples.

In the pathway analysis, the steroid hormone biosynthesis pathway had the highest number of hits. Accordingly, it has been hypothesized that the higher GC incidence rates in men than in women are due to the higher levels of sex steroid hormones [28]. Our results showed significant alterations in the levels of 11beta-ehydroxyandrost-4-ene-3,17-dione, adrenosterone, etiocholan-3α-ol-17-one 19-hydroxyandrost-4-ene-3,17-dione, 19-oxoandrost-4-ene-3,17-dione, and cortodoxone. The outcomes of this study regarding sex hormones showed mixed results; however, the levels of several metabolites from the corticosteroid pathway were consistently lower in GC patients. The level of aldosterone, which is included in the renin–angiotensin–aldosterone system (RAAS) as an adrenal component, was also lower in patients with GC in our study. The RAAS influences cell signaling, migration, death, and metastasis in cancer by balancing multiple receptor pathways [29]. Lee et al. reported that the use of RAAS inhibitors may be beneficial in gastrointestinal cancer prevention [30]. Busada et al. have suggested that glucocorticoids prevent gastric metaplasia by suppressing spontaneous inflammation [31]. They explored the role of glucocorticoids in the development of gastric inflammation and metaplasia. Our results suggest that glucocorticoids are essential for maintaining gastric homeostasis, and glucocorticoid deficiency may lead to GC development.

In tryptophan metabolism, the level of N-formylkynurenine was significantly higher in the GC group than in the IM group, while those of indole, L-tryptophan, indole acetaldehyde, indoleacetate, and L-kynurenine were significantly lower in the GC group than in the IM group. Tryptophan metabolism is altered in several cancer types [32], including GC [33]. L-tryptophan, an essential amino acid, has been studied in line with the kynurenine pathway, and its depletion is highly associated with cellular function and survival [34]. Tryptophan metabolism induces cancer progression via immunosuppressive responses; a recent study by Luo et al. concluded that tryptophan metabolism-associated genes could predict GC prognosis [34]. The level of indoleamine 2, 3-dioxygenase 1, which is a rate-limiting enzyme that converts tryptophan to kynurenine, is upregulated in multiple types of cancer, suggesting a possible role in carcinogenesis and as a potential biomarker [35]. In addition, the lower levels of indole, indole-acetaldehyde, and indoleacetate were observed in GC; indole metabolites could be formed in the stomach from tryptophan catabolism by the gut microbiota [36]. Moreover, several novel indole derivatives inhibit NEDDylation and MAPK pathways in GC cells [37].

Several metabolic alterations have been observed with purine derivatives. Cao et al. recently performed a pathway analysis using plasma metabolites and proposed that purine and arachidonic acid metabolism may play important roles in GC progression [38]. In our study, the GTP levels were significantly higher in patients with GC. Guanylate-binding protein 5, a member of the GTPase family, upregulates and promotes the proliferation and migration of GC cells, while the active GTP-bound form of Ras homolog family member A promotes tumorigenesis [39]. In addition, guanine nucleotide-binding protein subunit beta-4 plays a crucial role in the initiation and progression of cancers, including GC [40]. Purine derivatives, including inosine, hypoxanthine, and 5-aminoimidazole ribotide, are precursors of uric acid that are reduced in hyperuricemia. Uric acid plays an important role in carcinogenesis owing to its pro- and antioxidant properties, and high levels of uric acid are associated with cancer [41]. Our results support this finding; however, the connection between them remains unclear.

Several metabolites involved in arginine and proline metabolism, including S-adenosylmethioninamine, N-acetylputrescine, N_4_-acetylaminobutanal, N-carbamoyl sarcosine, arginine, creatine, sarcosine, N-methylhydantoin, and creatinine, were significantly altered in the GC group. Several studies have shown a correlation between abnormal amino acid metabolism and GC; arginine systemically decreased in GC, suggesting that it is a potential biomarker [42,43]. Creatine plays a key role in the recycling of adenosine triphosphate (ATP) [44], which is considered an energy currency for all cell behaviors, particularly in cancer cells, owing to its highly proliferative properties. In colorectal cancer cells, creatine is phosphorylated by creatine kinase, and phosphocreatine is utilized to provide energy for cell survival. Furthermore, creatine plays a role in macrophage polarization by inhibiting the M1-like phenotype and promoting the M2-like phenotype in macrophages [45]. The M2-like phenotype promotes metastasis and epithelial mesenchymal transition in GC cells [28]. Therefore, GC cells accelerate the physiological process leading to creatine depletion.

Several notable metabolites were identified through ANOVA testing with cancer stages, including N1, N2-diacetylspermine, creatine riboside, and N-formylkynurenine. Increasing levels of polyamines such as diacetylspermidine have been suggested as useful tumor markers for various cancer types, including GC [46]. Although not significant in the arginine and proline metabolic pathways in our study, creatine riboside is a cancer cell-derived metabolite that is associated with various cancer types, such as cervical, lung, and liver cancer, and has shown potential as a poor prognosis biomarker. This observation is consistent with the findings of the present study, in which the intensity of creatine riboside increased with increasing stage [47]. N-formylkynurenine was a significant metabolite in tryptophan metabolism between the GC and IM and showed higher intensity with increasing stage.

Figure 9 also shows that S-adenosylmethioninamine, methionyl-leucine, and S-(formylmethyl)glutathione stand out as promising candidates for the early detection of GC. S-adenosylmethioninamine is one of the significantly altered metabolites between GC and IM in the arginine and proline metabolism pathways from the KEGG pathway. S-adenosylmethioninamine, produced by the decarboxylation of S-adenosylmethionine, serves as a substrate necessary for the biosynthesis of polyamines such as spermine and spermidine. Polyamines are essential for the growth, differentiation, and development of eukaryotic cells. Therefore, an increase in S-adenosylmethioninamine levels promotes polyamine production, thereby promoting cancer progression [48]. S-(formylmethyl)glutathione is formed by the conjugation of bioactive compounds, for example, bromoacetaldehyde to glutathione, facilitated by glutathione S-(formylmethyl)transferase. Bromoacetaldehyde, a highly reactive compound, is derived from the carcinogen 1,2-dibromoethane [49]. Under conditions of glutathione deficiency, there is a possibility that the aldehyde intermediate, 2-bromoacetaldehyde, may interact with macromolecules, potentially contributing to cancer progression. When comparing IM and early GC, these differences were statistically significant (ANOVA *p* ≤ 0.05). However, as GC progresses, these differences tend to become more pronounced. Therefore, metabolites that show significant differences in the late stage may also become significant biomarkers for the early stage when validated in a larger cohort. Thus, we aim to explore the biomarker potential of these metabolites through further validation studies, leveraging existing knowledge to guide our selection and exploration process.

There are several studies on metabolic profiling in gastric cancer (GC), but direct comparisons with our study are challenging due to differences in study design and control samples. Hirayama et al. [20] obtained tumor and surrounding grossly normal-appearing tissues from 12 stomach cancer patients after surgical treatment, while our study involved 37 patients who underwent esophagogastroduodenoscopy for evaluation and biopsy of both intestinal metaplasia (IM) and GC lesions. Despite the differences in control tissue samples, the metabolic profiling of amino acids, particularly tryptophan and arginine metabolism, was consistent. Other studies have used varied methodologies and control samples, complicating direct comparisons. For instance, one study focused on nuclear magnetic resonance spectroscopy to determine macromolecules mainly in urine with some gastric tissues [21], while another utilized gas chromatography/mass spectrometry metabonomics to fingerprint tumor tissues and matched normal mucosae, identifying significant lipid metabolites [22]. Additionally, research on gastric cardia cancer employed proteomics and metabolomics to explore different metabolites in cancerous and non-cancerous tissues [23]. Lastly, a study examining metabolites in tissues and plasma from GC patients, postoperative GC patients, and control patients with chronic superficial gastritis (CSG) focused on the balance profile of metabolites between the tumor microenvironment and the systemic environment, further complicating direct comparisons with this study [24]. These variations highlight the complexity of comparing metabolic profiling results across different studies and emphasize the need for careful consideration of study design and control selection.

Our study has several limitations. Despite efforts to control for factors that could influence the metabolomics results, tissue specificity may have affected the concentrations of molecules. The heterogeneity of the study group and composition of the cancer tissue could not be avoided. The small sample size and limited ethnic diversity, primarily Koreans, also posed significant limitations. Expanding our study to include diverse ethnicities, healthy controls and a larger sample size could enhance its potential to identify new biomarkers for GC. There were several significant metabolites that could not be discussed due to the lack of literature describing them. To overcome these limitations, we would combine the study results with the gene variations using data from The Cancer Genome Atlas to provide more accurate and meaningful results in a further study.

## 4. Materials and Methods

### 4.1. Study Design and Patient Enrollment

The study protocol was reviewed and approved by the Institutional Review Board of Korea University Anam Hospital, Seoul, Republic of Korea (16 July 2018; 2018AN0265). Written informed consent was obtained from all patients prior to their participation in this study. This study was registered in the open registry of the Clinical Research Information Service (http://cris.nih.go.kr; KCT0005597 (accessed on 15 August 2024)). Forty patients with pathologically confirmed GC were recruited from Korea University Anam Hospital between August 2018 and December 2019. Written informed consent was obtained from all participants in accordance with the Declaration of Helsinki. The inclusion criteria were as follows: (1) pathologically diagnosed with GC and (2) aged between 20 and 80 years. The exclusion criteria were as follows: (1) previously treated for GC, (2) previously treated for *Helicobacter pylori* infection, and (3) with inability to discontinue gastrointestinal medication. Data on demographics, medical history, family history, medications, and laboratory tests were obtained. Accordingly, two patients with inadequate biopsy specimens and one patient whose IM were not confirmed during the pathologic test were excluded from the enrolled cohort. All three patients refused to undergo a restudy. Hence, a total of 37 patients were analyzed.

### 4.2. Endoscopic Sampling

All 37 patients underwent esophagogastroduodenoscopy for the evaluation and biopsy of both IM and GC lesions. Endoscopy was performed between 8 a.m. and 10 a.m. after more than 8 h of fasting. Gastric mucosal biopsies were performed using standard gastroscopic forceps to obtain samples measuring 1–2 mm. In each patient, 4 biopsy samples were collected, 2 from each of the pathology-confirmed GC lesions and another 2 from the IM lesions, at least 5 cm away from the cancer site. Tissues obtained from odd-numbered orders were sent to the pathology department for pathological confirmation, and tissues obtained from even-numbered orders were immediately frozen in liquid nitrogen and stored at −80 °C for further metabolite analysis. The samples were re-evaluated pathologically to ensure that the biopsies were performed in lesion areas where histological changes indicative of cancer and IM were present.

### 4.3. Quality Control Samples

A standard quality control (QC) strategy was employed to ensure reproducibility and stability. Initially, a pooled (QC) sample of all reconstituted tissue extracts was loaded and run more than ten times to condition the column before analyzing the study samples, as described previously [50]. Separate runs were conducted for QC samples of aqueous and organic extracts.

### 4.4. Sample Treatment for Metabolite Extraction

Tissue samples (5–10 mg) were added to pre-chilled 50% methanol (MeOH) to obtain aqueous metabolites. The volume of the solution was adjusted according to the weight of the sample starting with a maximum weight of 30 μL/mg. Tissue was sonicated and centrifuged at 13,000× *g* for 20 min, at 4 °C. Supernatant aliquots were transferred to Eppendorf tubes. Samples were spun in a vacuum concentrator for 3 h at 45 °C until dry, and stored at −40 °C until analysis. The aqueous pellet was redissolved in 60 μL of solvent mixture of acetonitrile (ACN) and water in 95:5 ratio and transferred into a new Eppendorf tube after centrifugation for 20 min at 13,000× *g* at 4 °C. 

Organic extracts were harvested from the residual pellets of the aqueous extract. Following aqueous extraction, a solution of pre-chilled dichloromethane/MeOH (3:1) was added to the residual pellet. The volume of the solution was proportional to the sample weight (as described in the previous paragraph; aqueous extraction). The samples were then centrifuged at 13,000× *g* for 20 min, and the organic phase supernatant was aliquoted into glass vials. The samples were allowed to evaporate overnight at room temperature in an extractor hood and stored at −40 °C until analysis. The dry residue was redissolved in 60 μL of water/ACN/isopropanol (1:1:2). The supernatant was centrifuged (5000× *g*, 4 °C, 10 min), as previously described [51]. 

### 4.5. Analysis of Metabolites by Liquid Chromatography with Tandem Mass Spectrometry

An ultra-performance liquid chromatography system (UPLC; Agilent 1260 Infinity Quaternary, Santa Clara, CA, USA) coupled with an Agilent liquid chromatography with tandem mass spectrometry (LC-MS/MS) was performed using a Q-TOF 6550 iFunnel Q-TOF mass spectrometer (Agilent) for metabolomic profiling. The samples were analyzed using C18 Synchronis aQ (1.9 μm, 100 × 2.1 mm; Thermo Fisher Scientific, Inc., Waltham, MA, USA). The autosampler and column temperatures were maintained at 10 °C and 45 °C, respectively. Solvent A comprised 0.1% formic acid in water (HPLC grade, Tedia, OH, USA), while solvent B comprised 0.1% formic acid in acetonitrile (HPLC grade, Tedia, OH, USA). The injection volume and flow rate were 5 µL/min and 0.4 mL/min, respectively. The HPLC gradient was programmed as follows: 95% water for 1 min, a linear decrease to 55% water over 8 min, a descending gradient to 10% water over 3 min, hold for 1.5 min, and a return to 95% water over 0.1 min. The electrospray ionization detector was operated with a curtain gas of 35 psi at 250 °C, supplied at 14 mL/min, and a sheath gas temperature of 250 °C, supplied at a flow rate of 11 mL/. Ion detection was set from 50 to 1000 with a resolution of 20,000 over 15 min. All samples were run in triplicate, and data for each ionization technique were acquired in the positive ion mode.

### 4.6. Alteration of Metabolite Intensities Caused by GC Corrected with Stage

Multivariate analysis was conducted using apLCMS within an ion range of 50 to 1000 based on the mass spectral data [52], the R package xMSanalyzer [53], and xmsPANDA [54,55] for aqueous and organic extracts. In the Manhattan plot, the y-axis represents the negative log of the *p*-values, while the x-axis represents *m*/*z*. Significant metabolites were selected using an FDR q-value of ≤0.05 with the Manhattan plot. Prior to model fitting, the features were normalized to the median, log_2_ transformed, and Pareto scaled values. Unsupervised PCA and supervised sPLS-DA were performed to visualize the metabolic differences between the IM and GC. In the sPLS-DA, the first component of the variable importance in projection (VIP) value > 1.5 from sPLS-DA was considered influential for the separation of samples. The pathway analysis was considered significant if five or more *m*/*z* values were changed. The annotation of *m*/*z* values was matched to metabolites from mass spectrometry databases, including xMSannotator, Human Metabolome Database, and KEGG [56,57,58,59,60]. MetaboAnalyst 5.0 software (www.metaboanalyst.ca) was used to determine the influence of these metabolites on the carcinogenic transformation from IM to GC. The number of pathway hits was calculated, and four or more hits were considered significant. 

## 5. Conclusions

In this study, we successfully identified significant metabolic differences between IM and GC using a comprehensive metabolomic approach. The steroid hormone biosynthesis, tryptophan metabolism, purine metabolism, and arginine and proline metabolism were altered significantly. Aldosterone, N-formylkynurenine, GTP, arginine, S-adenosylmethioninamine, and creatine riboside showed distinct alterations, highlighting their potential as diagnostic biomarkers for GC. Additionally, the correlation of certain metabolite intensities with the stage of GC progression suggests their utility not only in early diagnosis but also in monitoring disease advancement.

These findings provide valuable insights into the metabolic alterations associated with gastric carcinogenesis and offer promising biomarkers for the early detection and prognosis of GC. While our current study utilized invasive endoscopic sampling to identify these key metabolites, subsequent studies should aim to validate these biomarkers using noninvasive methods such as blood or urine tests. Furthermore, validation in larger and more diverse cohorts is essential to explore their potential as noninvasive diagnostic tools. The identification and application of these biomarkers could improve patient outcomes through early detection and targeted therapeutic strategies.

## Figures and Tables

**Figure 1 ijms-25-09509-f001:**
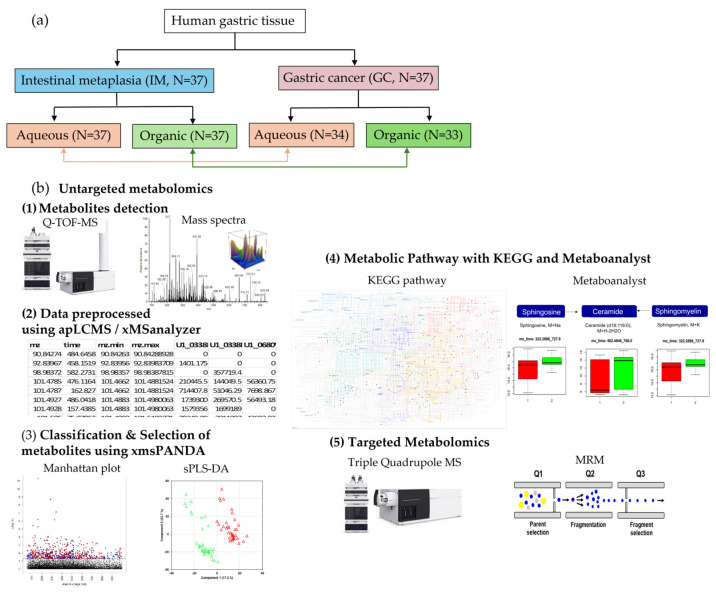
Schematic overview of the biomarker identification criteria applied in this study. (**a**) The metabolic overview started with the extraction of metabolites from 37 gastric cancer (GC) and intestinal metaplasia (IM) tissues. (**b**) The scheme of untargeted metabolomics analysis. (1) Metabolites detection was done using Q-TOF-MS, generating mass spectra. (2) Data preprocessed using apLCMS version 6.3.8 and xMSanalyzer version 2.0.6.1. (3) Metabolic profiling with Manhattan plot with false discovery rate (FDR) analysis and sparse partial least square discriminant analysis (sPLS-DA). (4) Pathway analysis with Kyoto Encyclopedia of Genes and Genomes (KEGG). (5) Biomarker quantification using multiple reaction monitoring (MRM). apLCMS: adaptive processing of liquid chromatography-mass spectrometry. FDR, false discovery rate; GC, gastric cancer; IM, intestinal metaplasia; KEGG, Kyoto Encyclopedia of Genes and Genomes; MS, mass spectrometery; Q-TOF-MS, quadrupole time-of-flight mass spectrometry; sPLS-DA: sparse partial least square discriminant analysis.

**Figure 2 ijms-25-09509-f002:**
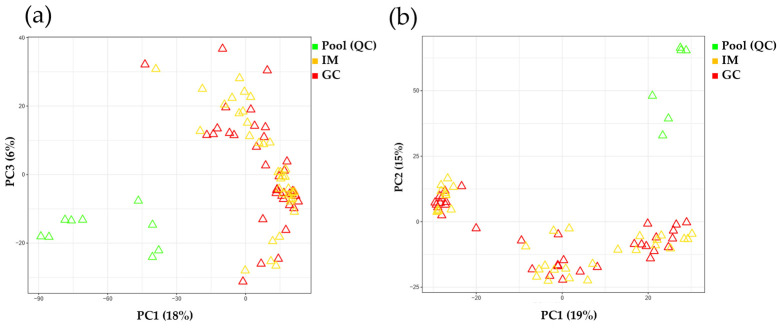
Analytical performance evaluation by comparing pool and case samples. (**a**) PCA using 665 significant features (FDR q ≤ 0.05) from aqueous extraction data. (**b**) PCA using 278 significant features (FDR q ≤ 0.05) from organic extraction data. FDR, false discovery rate; GC, gastric cancer; IM, intestinal metaplasia; PCA, principal component analysis; PC1, principal component 1; PC2, principal component 2; PC3, principal component 3; QC, quality control.

**Figure 3 ijms-25-09509-f003:**
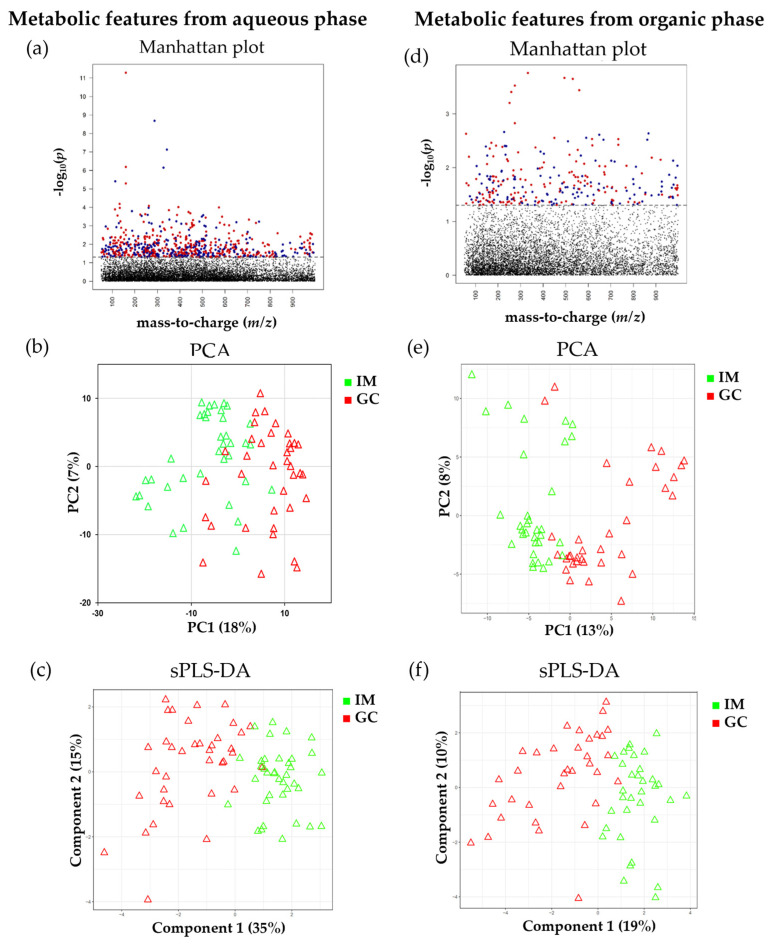
Manhattan plot, PCA, and sPLS-DA between IM and GC. The Manhattan plot presents the significant features (FDR q ≤ 0.05) as colored dots, while their distribution is expressed in *m*/*z*. (**a**) Manhattan plot showing 665 significant features (FDR q ≤ 0.05) derived from the aqueous data. (**d**) Manhattan plot with 278 significant features (FDR q ≤ 0.05) derived from the organic data. PCA shows the separation of samples: (**b**) aqueous data and (**e**) organic data. sPLS-DA shows the separation of samples: (**c**) aqueous data and (**f**) organic data. PCA, principal component analysis; PC1, principal component 1; PC2, principal component 2, sPLS-DA, sparse partial least squares discriminant analysis.

**Figure 4 ijms-25-09509-f004:**
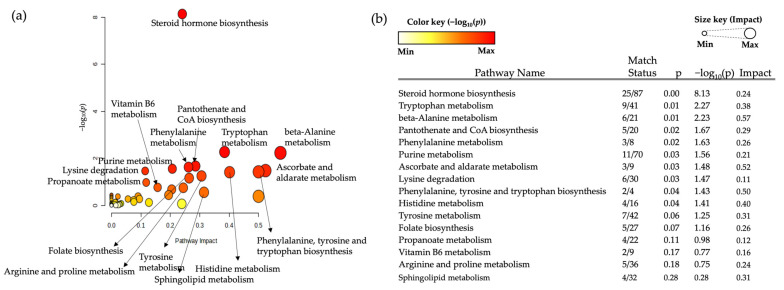
Overview of the pathway analysis of the significant metabolites extracted from the combined aqueous and organic phases. (**a**) The bubble plot shows the pathways by impact (x-axis) and −log_10_(*p*) (y-axis). The color and size of each bubble represent the −log_10_(*p*) and impact, represented as color and size keys, respectively. (**b**) The top 16 pathways based on the −log_10_(*p*) are listed alongside their match status, indicating the hit metabolites to whole metabolites involved in each pathway, *p*-value, and impact.

**Figure 5 ijms-25-09509-f005:**
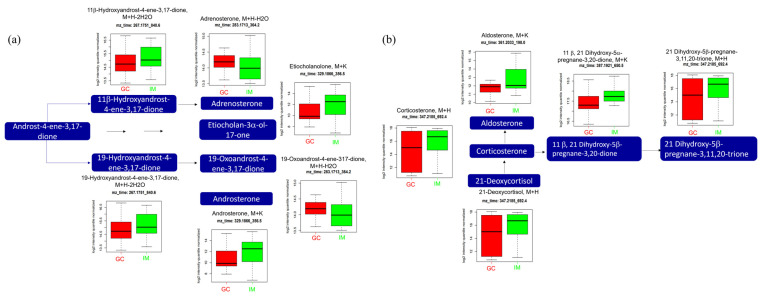
Analysis of significantly altered metabolites between GC and IM in the steroid hormone biosynthesis pathway from the KEGG pathway. (**a**) Pathway of androgen steroid and (**b**) pathway of mineralocorticoid. All metabolites in the figure a and b were significantly altered (FDR, q ≤ 0.05). Boxplots illustrate the upper quartile, median (dashed bar), and lower quartile, with whiskers indicating the maximum and minimum values. FDR, false discovery rate; GC, gastric cancer; IM, intestinal metaplasia; KEGG, Kyoto Encyclopedia of Genes and Genomes.

**Figure 6 ijms-25-09509-f006:**
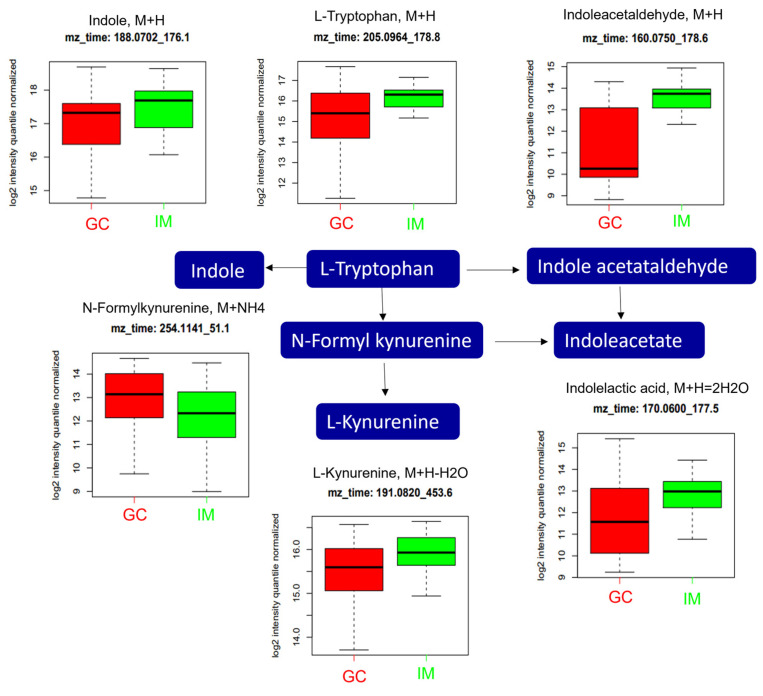
Analysis of significantly altered metabolites between GC and IM in the tryptophan metabolism pathway from the KEGG pathway. All metabolites in the figure were significantly altered (FDR, q ≤ 0.05). Boxplots illustrate the upper quartile, median (dashed bar), and lower quartile, with whiskers indicating the maximum and minimum values. FDR, false discovery rate; GC, gastric cancer; IM, intestinal metaplasia; KEGG, Kyoto Encyclopedia of Genes and Genomes.

**Figure 7 ijms-25-09509-f007:**
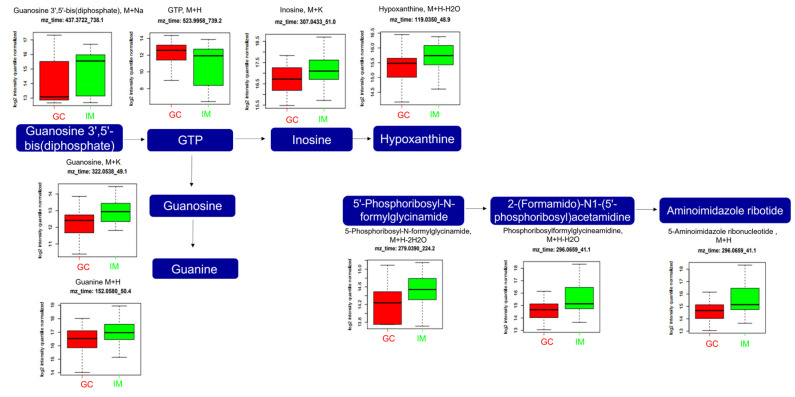
Analysis of significantly altered metabolites between GC and IM in the purine metabolism pathway from the KEGG pathway. All metabolites in the figure were significantly altered (FDR, q ≤ 0.05). Boxplots illustrate the upper quartile, median (dashed bar), and lower quartile, with whiskers indicating the maximum and minimum values. FDR, false discovery rate; GC, gastric cancer; GTP, guanosine triphosphate; IM, intestinal metaplasia; KEGG, Kyoto Encyclopedia of Genes and Genomes.

**Figure 8 ijms-25-09509-f008:**
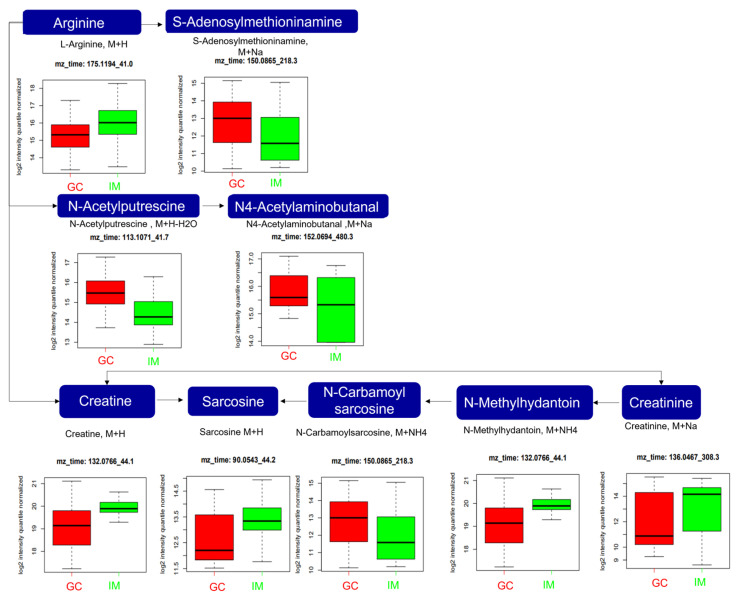
Analysis of significantly altered metabolites between GC and IM in the arginine and proline metabolism pathways from the KEGG pathway. All metabolites in the figure were significantly altered (FDR, q ≤ 0.05). Boxplots illustrate the upper quartile, median (dashed bar), and lower quartile, with whiskers indicating the maximum and minimum values. FDR, false discovery rate; GC, gastric cancer; IM, intestinal metaplasia; KEGG, Kyoto Encyclopedia of Genes and Genomes.

**Figure 9 ijms-25-09509-f009:**
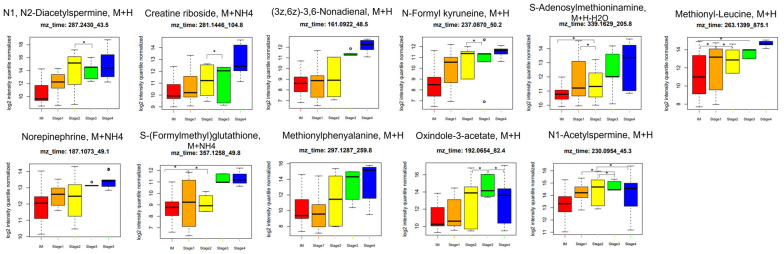
Relative metabolite intensities in tissues across various stages of GC at diagnosis. Boxplots illustrate the upper quartile, median (dashed bar), and lower quartile, with whiskers indicating the maximum and minimum values. Relative intensities of significant compounds correlated with four stages of GC. * *p* ≤ 0.05. GC, gastric cancer.

**Table 1 ijms-25-09509-t001:** Clinical characteristics of the participants.

Variables	Total (*n* = 37)
Age, median	62.5 ± 13.9
Sex, *n* (%)	
Male	23 (62.2)
Female	14 (37.8)
Location, *n* (%)	
Upper	5 (13.5)
Middle	11 (29.7)
Lower	21 (56.8)
Stage, *n* (%)	
I	15 (40.5)
II	7 (18.9)
III	4 (10.8)
IV	11 (29.7)
Pathologic differentiation	
Poorly	22 (59.5)
Moderately	13 (35.1)
Not applicable	2 (5.4)
Lauren classification	
Intestinal	17 (45.9)
Diffuse	12 (32.4)
Mixed	8 (21.6)
Current *Helicobacter pylori* infection	26 (70.3)
Family history of gastric cancer	5 (13.5)

## Data Availability

The data presented in this study are available upon request from the corresponding author.

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
