# Peer review of "Gastric Cancer and Intestinal Metaplasia: Differential Metabolic Landscapes and New Pathways to Diagnosis"

_ijms, 2024, doi:10.3390/ijms25179509_

Round 1

Reviewer 1 Report

Comments and Suggestions for Authors

This study compared the metabolic profiles of intestinal metaplasia and gastric cancer and to understand the pathways involved in gastric carcinogenesis. Using advanced analytical techniques, they analyzed samples from 37 patients and identified significant metabolic differences between IM and GC tissues. The findings show the potential for developing new diagnostic tools for GC patients. This is a well-designed study. I have a few points to make.

1. There were significantly increased metabolite intensities observed in the late stage of GC, suggesting that they may not be suitable for early detection/prediction of GC.

2. It would be interesting to know whether these metabolites can be found in blood samples, as the blood samples are easier to obtain than tissue samples.

3. The legend of Figure 3d indicates that there were 665 features from organic data, which is not consistent with the information provided in Results 2.2.

Reviewer 2 Report

Comments and Suggestions for Authors

A metabolomics approach is used in this study with the aim of finding biomarkers to distinguish between intestinal metaplasia (IM) and gastric cancer (GC). A number of metabolites related to diverse metabolic pathways are reported as potential biomarkers.

The study is conducted using an UPLC-MS platform according to established standards in the field.

A number of issues are pointed out to be addressed by the authors:

1) It is not clear how many samples are from IM and from GC when speaking of 37 patients.

2) In order to know how the current state of IM is physiologically defined it would be necessary to have included healthy controls in the study.

3) In table 1 it is indicated that 26 patients had Helicobacter pylori infection but it is also indicated that Helicobacter pylori treatment was consider an exclusion condition.

4) Separation is not good even in PLS-DA, with very low variability explanation by both components

5) Actually 665 plus 278 features were matched in KEGG? And, all of them were submitted to pathway analysis in MetaboAnalyst?

6) Improvement of figure 5 quality would be welcome

7) Give p values of t-test where appropriate

Reviewer 3 Report

Comments and Suggestions for Authors

In this manuscript, the authors aimed to discover new biomarkers using global metabolomics for early detection and prognosis of gastric cancer (GC), which may overcome some limitations of the current approach of endoscopy.  Different from existing studies, the work reported here focused on metabolic changes between IM (intestinal metaplasia) and GC tissues for the first time.  Some metabolic pathways and metabolites were identified as potential biomarkers for further validation in the future. The manuscript is organized and written well.  It is easy to follow.  The conclusion is largely supported by the experimental results.  Some comments are as follows:

1.  In the discussion section, it would be helpful for the authors to compare their work with the existing work from references 20-24 because they are closely related.

2.  In the conclusion section, the authors claimed their metabolic method as noninvasive.  However, it still relies on an endoscope for sampling based on the method section.  Could the authors provide some further explanation as to why they would consider their method as noninvasive?

3.  In figures 5-9, it would be helpful to add more explanation details in the legend, e.g. the dashed bars, the colored boxes and the bold lines.

Minor points:

1.  In figure 1b, it would be helpful to add 1, 2, 3, and 4 to match with the legend.

Round 2

Reviewer 2 Report

Comments and Suggestions for Authors

Certainly it would be great to improve figure quality but I understand constrains pointed out by the authors in their answers

Author Response

Thank you for your valuable comments.

We greatly appreciate your feedback, which has significantly enhanced the quality of our paper.

As per your suggestion, we have made efforts to improve the figure quality. However, it has been quite challenging to balance maintaining an appropriate size for the figure while ensuring clarity and legibility, particularly due to the large amount of text and graphs included.

We revised Figure 1-4 again, unified the fonts, increased the font size, and made minor adjustments to improve readability. 

Thank you again for your thoughtful review and understanding